# Highly Regioselective Synthesis of Bisadduct[C_70_] Additive toward the Enhanced Performance of Perovskite Solar Cells

**DOI:** 10.3390/nano12142355

**Published:** 2022-07-09

**Authors:** Muqing Chen, Yanyan Zeng, Gui Chen, Yongfu Qiu

**Affiliations:** School of Environment and Civil Engineering, Dongguan University of Technology, Dongguan 523808, China; double124@126.com (Y.Z.); chengui@dgut.edu.cn (G.C.)

**Keywords:** fullerene, C_70_, bisadduct, regioselectivity, rigid tether, perovskite solar cells, additive

## Abstract

The high-regioselective synthesis of bisadducts based on low-symmetry C_70_ has been a challenging work due to the large amount of formed regioisomers, which require tedious separation procedures for isomeric purity and block their application in different fields. Herein, we successfully obtained a novel 1, 2, 3, 4-bis(triazolino)fullerene[C_70_] **2** with high regioselectivity by the rigid tether-directed regioselective synthesis strategy and the corresponding molecular structure was unambiguously confirmed by single-crystal X-ray crystallography characterization. The crystal data clearly show that the addition occurs at the domain of corannulene moiety at the end of ellipse C_70_ as well as the 1, 2, 3, 4-addition sites located at one hexagonal ring with a [6,6]-closed addition pattern. Furthermore, 2 was applied as an additive of perovskite layer to construct MAPbI3-based regular (n-i-p) perovskite solar cells, affording the power conversion efficiency (PCE) of 18.59%, which is a 7% enhancement relative to that of control devices without additive.

## 1. Introduction

The functionalization of fullerenes via versatile chemical reactions has been evidenced as an effective strategy to improve their properties for further application in the field of materials science and energy conversion devices [1,2,3,4,5]. The monoadducts of fullerenes, especially [6,6]-phenyl-C_61_-butyric acid methyl ester (PCBM), have been widely used as electron transport layers and additives within perovskite solar cells (PSCs), effectively improving the device performance [6,7]. Relative to the monoadducts of fullerenes, the corresponding bisadducts as additives have fewer reports, but have exhibited superior performance compared to that of their corresponding monoadduct counterparts [8]. In 2017, Grätzel and co-workers reported that isomer-pure bis-PC_61_BM (α-bis-PC_61_BM) was incorporated into CH_3_NH_3_PbI_3_ (MAPbI_3_) perovskite film through an anti-solvent approach, and revealed that α-bisPC_61_BM plays the templating agent role in enhancing the crystallinity of perovskite films, delivering a PCE of 20.8% for n-i-p bulk heterojunction (BHJ)-PSCs, which is superior to that of devices with PCBM as the additive (19.9%) [8].

Due to the larger π-electron system and distinct charge distribution of the C_70_ cage relative to that of C_60_, the corresponding C_70_-based bisadducts were expected to become promising additives applied within PSCs. Relative to C_60_ bearing only one type of [6,6]-bond, however, C70 with D5h symmetry has four inequivalent types of [6,6]-bonds defined as α, β, γ, and δ (Figure 1), leading to the C_70_-based bisadducts having 38 possible isomers on the condition of identical groups and additions occurring exclusively on [6,6]-bonds [9,10,11]. More importantly, some regioisomerically pure bisadducts were reported to have a better performance in organic photovoltaic solar cells than that of the isomeric mixtures [12,13,14,15,16]. This renders us a hint that the isomeric pure bisadducts [C_70_] applied within PSCs for improving device performance seem more promising than those of mixed isomers.

The tether-directed remote functionalization reported firstly by Diederich et al. provided an effective approach to reduce the isomers of bisadduct [C_70_], which is beneficial to greatly improve the yield of bisadducts [C_70_] as well as to reduce the difficulty of separating the deteriorated bisadduct isomers [9,10,11,17,18,19]. Luis Echegoyen et al. reported the regioselective synthesis of easily isolable bismethano-derivatives of C_70_ through short tethered bis-*p*-toluenesulfonyl hydrazone as the addend precursor, affording only two bisadduct isomers [9]. It is obvious that a suitable and short tether connecting two functional groups enables the decreased bisadduct [C_70_] isomers formation. Recently, we employed the rigid phenyl tether azide to functionalize C_60_, affording a bisadduct [C_60_] with a high regioselectivity [20]. Considering the merit of the short and rigid azide phenyl tether on decreasing the regioisomers, whether this method can be extended to C_70_ to form bisadduct [C_70_] with a high regioselectivity and act as an efficient additive within perovskite to improve the performance of PSCs is an interesting issue. 

Herein, we reported the high regioselectivity synthesis of a novel bisadduct [C_70_] isomer **2** using a short and rigid azide phenyl tether strategy. Importantly, the crystal structure of **2** was unambiguously confirmed by single-crystal X-ray crystallography, showing that four addition sites occur at a hexagonal ring with a [6,6]-closed addition pattern. The highly regioselective formation of 2 was attributed to the collaborative contribution of the high-active α [6,6]-bonds and short rigid tether approach. Furthermore, the bisadduct 2 was applied as an additive to perovskite film to construct bulk heterojunction MAPbI_3_-based PSCs, delivering the champion PCE of 18.59%, which was attributed to the improved crystal quality of the perovskite upon the incorporation of **2**.

## 2. Experiment

### 2.1. Characterization Techniques

C_70_ was synthesized with an improved direct-current arc discharge method and isolated with high-performance liquid chromatography (HPLC), conducted on an LC-9130 NEXT machine (Japan Analytical Industry Co., Ltd, Akishima, Japan) with toluene as the mobile phase. Matrix-assisted laser desorption/ionization time-of-flight (MALDI-TOF) mass spectrometry measurements were performed on a BIFLEX III spectrometer (Bruker Daltonics Inc., Billerica, MA, USA) using 1, 1, 4, 4-tetraphenyl-1, 3-butadiene as the matrix. UV-vis spectra were obtained from a PE Lambda 750S spectrometer (PerkinElmer, Waltham, MA, USA) in toluene. Electrochemical measurements were carried out under argon atmosphere with tetra-n-butylammonium hexafluorophosphate (0.1 M) in o-DCB using a computer-controlled CHI660C electrochemical workstation with a platinum disc as the working electrode, a platinum-wire as the auxiliary electrode, and Ag wire as the reference electrode. Potentials were referenced to the ferrocenium/ferrocene (FeCp^2+/0^) couple by using ferrocene as an internal standard. Scanning electron microscope (SEM) images were acquired with a field-emission scanning electron microscope (FEI Quanta 200). Atomic force microscope (AFM) images were recorded on an XE-7 scanning probe microscope in non-contact mode (Park systems, Suwon, Korea).

### 2.2. The Synthesis of 1, 2-Bis(Azidomethyl)-Benzene

1, 2-bis(bromomethyl)benzene (1 g, 3.79 mmol) and sodium azide (0.59 g, 9.09 mmol) were added into the mixed solution of deionized water (10 mL) and ethyl acetate (15 mL). After the solution become transparent by stirring, the reaction temperature was heated up to 75 °C overnight. The reaction was monitored by TLC, and then the organic layer was separated and dried by anhydrous magnesium. The obtained product named as 1, 2-bis(azidomethyl)-benzene was colorless oil (0.7 g, 99%). ^1^H NMR (600 MHz, CDCl_3_) (δ 7.46–7.36 (m, 4H), 4.46 (s, 4H)) and ^13^C NMR (151 MHz, CDCl_3_) (δ 133.88, 130.17, 129.05, 52.27).

### 2.3. The Synthesis of 2

1, 2-bis(azidomethyl)-benzene (62 mg, 0.3 mmol) was added to the solution of C_70_ (84 mg, 0.1 mmol) in CB (35 mL). The mixture was allowed to stir at 75 °C. The solution was monitored by analytical HPLC. After the reaction was terminated, the pure product **2** was obtained as brown-black powder (44.9 mg, 43.7%) by preparative HPLC.

### 2.4. Crystal Growth and Measurements

Black single crystals of **2** suitable for single-crystal X-ray diffraction were prepared by slowly diffusing n-hexane into the CS_2_ solution of **2** at room temperature for four weeks. Crystal data of **2** were collected with the synchrotron radiation (λ = 0.7749 Å) at Beamline 11.3.1 at the Advanced Light Source, Lawrence Berkeley National Laboratory. The data were reduced utilizing Bruker SAINT, and a multi-scan absorption correction was applied using SADABS. The structure was solved with the direct method and refinement with SHELXL-2014/7 [21]. Crystal data of **2** were stored in the Cambridge Crystallographic Data Centre (CCDC,1403971). 

## 3. Results and Discussion

The synthesis of bisadduct [C_70_] **2** is shown in Figure 1. A mixture of C_70_ and 1, 2-bis(azidomethyl)benzene in chlorobenzene (CB) was heated at 75 °C under argon atmosphere. The reaction process was monitored with analytical HPLC (Appendix A in Appendix A. Before the reaction started, the mixed solution only had one peak assigned to C_70_ in Appendix A. When the reaction prolonged to 4 h, a new peak at 7.8 min was assigned to the bisadduct **2**. Continuing the reaction time, the peak intensity of **2** continued to increase along with the decrease of pristine C_70_. After the trace multi-adduct byproduct appeared at 5 min in HPLC profile, the reaction was terminated. Interestingly, only one bisadduct [C_70_] **2** without any isomers was obtained in this reaction, indicating the high regioselectivity of this reaction via the rigid tether strategy. Finally, the reaction mixture was subjected to prepared HPLC, yielding 43.7% of bisadduct **2** based on the consumed C_70_.

Importantly, the molecular structure of **2** was unambiguously confirmed by single-crystal X-ray diffraction measurements (Figure 2, Appendix A). Black crystals suitable for single-crystal measurements were obtained by slow diffusion of n-hexane into the carbon disulfide solution of **2** over a period of four weeks at room temperature. It is clearly shown that the four addition sites locate at a hexagonal ring with an α-1-α [6,6]-closed addition pattern. The bond lengths of C1A-C4A and C2A-C3A are 1.575(7) Å and 1.571(8) Å, respectively, which fall in the range of a single bond, confirming the [6,6]-closed addition pattern (Figure 2a,b). The bond lengths of N2A-N3A (1.245(7) Å) and N5A-N6A (1.254(6) Å) are in the range of the N=N double bond versus the single bond of N1A-N2A (1.343(7) Å) and N4A-N5A (1.335(5) Å), confirming the formation of triazolino rings. The crystal packing shows that the rigid phenyl ring locates between two C_70_ carbon cages with the intermolecular distance of 3.43 Å, indicating the strong π–π interaction between molecules, which benefits the electron transport among these molecules (Figure 2c). 

The electronic configuration of **2** was investigated by UV-vis spectroscopy as illustrated in Figure 3a. Compared to pristine C_70_ with absorption peaks at 334 nm, 363 nm, and 382 nm as well as the wide absorption region in the range of 415–619 nm, **2** exhibits a relatively smooth absorption band in the range of 300–700 nm, demonstrating that the electronic structure of **2** is completely different from that of C_70_ via the functionalization of rigid phenyl tether azide.

Redox properties of **2** were studied using cyclic voltammetry (CV) measurement at a scan rate of 100 mV s^−1^ in *o*-dichlorobenzene (ODCB) (Figure 3b). The corresponding redox data are summarized in Appendix A. The CV curve of **2** shows three pairs of reversible reduction peaks which cathodically shifted about 0.02 mV for the first and second redox peaks and 0.22 mV for the third redox peak compared to those of C_70_, which is similar to the results of C_70_-bis-anthracene reported by Echegoyen et al. [22]. The shifts were attributed to the introduction of the bis(triazolino) moiety enabling the bond saturation of C_70_ by chemical functionalization [23]. 

In order to understand the highly regioselective formation of bisadduct [C_70_] **2**, we proposed a possible reaction mechanism. Among all the types of C-C bonds, including α, β, γ, and δ, within C_70_, α-bonds at the poles of the molecule are the most reactive, therefore becoming the most favored sites for additions [22]. According to the reactivity discrepancy of the C-C bonds, one of the azide groups within the rigid azide phenyl tether first attracts the α C-bond C bond to graft a triazolino moiety onto the carbon cage. Later, another azide group has the possibility to attack the adjacent β, γ, and δ bonds, which is highly dependent on the tether properties including the length and rigidity of the tether (Figure 2). The selective addition toward the adjacent α C-C bond was attributed to the short and rigid phenyl tether having the only chance to contact the adjacent reactive α C-C bond, forming the α-1-α adduct with a high regioselectivity. 

To evaluate the function of additive **2** within the MAPbI_3_ layer, we fabricated regular-structure (n-i-p) PSC devices with the structure of ITO/SnO_2_/MAPbI_3_+**2**/spiro-OMeTAD/Au (Figure 4a). Additive **2** was firstly dissolved in chlorobenzene (CB) solvent and was doped into the perovskite precursor solution with different doping ratios (0.025 wt%, 0.05 wt%, 0.1 wt%) to determine the optimized additive concentration. To preclude the influence of the CB solvent on the device performance, we added the same volume amount of CB into the perovskite layer to construct the devices (0 wt% additive). Current density–voltage (J–V) curves of the PSC devices with and without additive **2** under 1 sun illumination are compared in Figure 4b. The corresponding photovoltaic parameters, including open-circuit voltage (V_oc_), short-circuit current (J_sc_), fill factor (FF), PCE, series resistance (R_s_), and shunt resistance (R_sh_), are summarized in Table 1. The control device without additive shows an average PCE of 16.99% with a V_oc_ of 1.07 V, a J_sc_ of 22.15 mA cm^−2^, and an FF of 71.41%. Meanwhile, the PSCs with absolute CB solvent have a PCE of 17.07%, which is similar to that of the control devices, indicating that the CB solvent has a negligible effect on the device performance. When additive **2** was incorporated into the perovskite film with a relatively low ratio (0.025–0.1 wt%), the device performance of the PSCs had an obvious alteration, especially for the optimized doping ratio of 0.05 wt%, affording an average PCE of 17.65% obtained from a V_oc_ of 1.09 V, a J_sc_ of 22.31 mA cm^−2^, and an FF of 78.26%. Meanwhile, the champion device with a PCE of 18.59% had an approximately 7% enhancement compared to that of the control device. The doping ratio of additive **2** continued to be increased up to 0.1 wt%; however, the device performance obviously decreased. This can be attributed to the higher additive incorporation within perovskite perhaps resulting in excess nucleation sites, leading to the smaller grain size and more grain boundary formation. The variation tendency of PCE as well as other photovoltaic parameters was conformed further by comparing the statistical photovoltaic data obtained for 14 independent devices (Appendix A shows box plots of the photovoltaic parameters). Among the three photovoltaic parameters, the PCE enhancement after the incorporation of additive **2** primarily resulted from the increase of Jsc (from 22.15 to 22.20 mA cm^−2^, a ~0.2% enhancement), and FF (from 71.41% to 73.21%, a ~2.5% enhancement).

Pristine MAPbI_3_ perovskite has been reported to take homogeneous nucleation process followed by the growth of the perovskite seed crystal [24]. After the incorporation of additive **2** into the perovskite precursor, it plays the role of the heterogeneous nucleation sites for perovskite crystallization, which enables a much lower nucleation barrier relative to that of the homogeneous nucleation process because of the decreased Gibbs free energy, facilitating the formation of a large grain size of perovskite [25,26]. SEM measurements were employed to verify the change of the grain size of the perovskite films after the incorporation of additive **2**. The control perovskite films without additive **2** show a morphology with a relatively small grain size centered at ca. 193 nm. Interestingly, when 0.05 wt% **2** was incorporated, the average perovskite grain size increased obviously up to 370 nm (Figure 5), confirming that function of **2** on promoting the grain growth. The enlarged grain size was beneficial to reduce the grain boundary of the perovskite film, leading to the decreased defects/trap-states at the grain boundary, which therefore contributed to the enhanced performance of the PSCs. Meanwhile, AFM measurements were performed to monitor the surface morphologies of the perovskite films with and without additive **2** (Appendix A. The roughness (25.17 nm) of the perovskite films with additive **2** (0.05 wt%) was larger than that (12.45 nm) of the control perovskite film, which is attributed to the aggregation of fullerene derivative **2**, similar to what was reported for a C_60_-PyP additive [26].

Meanwhile, to verify the interaction between the perovskite film and additive **2**, X-ray photoelectron spectroscopy (XPS) was performed as shown in Appendix A. The pristine perovskite film shows the characteristic Pb 4f_5/2_ and Pb 4f_7/2_ signals at 143.26 and 138.40 eV as well as the I 3d_3/2_ and I 3d_5/2_ signals at 630.89 and 619.41 eV, respectively. After the introduction of additive **2** into the perovskite film, the corresponding Pb 4f_7/2_ and Pb 4f_5/2_ signals exhibit the shift of 0.07 eV toward the lower core-level binding energies, indicating the relatively weak interaction between Pb^2+^ ions and additive **2**. As for the I 3d_3/2_ and I 3d_5/2_ signals, the perovskite films with additive **2** show the signals at 630.72 and 619.26 eV, exhibiting more obviously negative shifts with the value of 0.17 eV compared to those of the pristine films. This stronger interaction between additive **2** and I-ions indicate the efficient passivation toward PbI_3_-antisite defects during the perovskite self-assembly [27].

## 4. Conclusions

In summary, we performed a highly regioselective reaction of C_70_ with 1, 2-bis(azidomethyl)benzene under simple thermal conditions by the rigid short-tether strategy, affording only bisadduct **2**. The molecular structure of **2** was unambiguously confirmed by single-crystal X-ray diffraction measurements, showing that two triazolino moieties were grafted onto the carbon cage of C70 with an α-1-α [6,6]-closed addition pattern within one hexagonal ring. The formation mechanism of 2 shows that the highly regioselective formation of 2 is attributed to the collaborative influence of the short and rigid phenyl tether and highly active [6,6]-bonds. Thereafter, 2 was employed as the additive to MAPbI3 perovskite to construct n-i-p bulk heterojunction PSCs, delivering the best PCE of 18.59%, which is a 7% enhancement compared to that of control devices without additive. The performance improvement of the PSCs after the incorporation of **2** is because **2** acted as heterogeneous nucleation sites to enlarge the grain size of perovskite and decrease the grain boundary defects of the perovskite layer, contributing the elevated PCE of the PSCs.

## Data Availability

The data presented in this study are available upon request from the corresponding author. Meanwhile, the detailed crystal data can be obtained free of charge from The Cambridge Crystallographic Data Centre via www.ccdc.cam.ac.uk/data_request/cif (accessed on 30 January 2020).

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
