# Peer review of "Highly Regioselective Synthesis of Bisadduct[C70] Additive toward the Enhanced Performance of Perovskite Solar Cells"

_nanomaterials, 2022, doi:10.3390/nano12142355_

Round 1

Reviewer 1 Report

The authors present their results on the successful synthesis of a C70 based additive to increase the power conversion efficiency of hybrid perovskite based solar cells. The investigation seems to be sound and the results and discussion are presented well. Apart from some smaller issues listed below I can support the publication in Nanomaterials.

1. Language

The whole manuscript would benefit from some language editing. At the moment, some of the sentences are misleading, making it harder to understand the content of the manuscript. Just as an example, on page 1, line 24, please replace “as the effective strategies” by “as effective strategies”. Further down, line 27, I assume you meant “widely used as electron transport layer”? Again, further down on line 29, the sentence “whereas which have exhibited superior performance” makes no sense at the moment. There are more instances like these, so please have a careful look at the whole manuscript again.

2. Figures

In figure 4 (a), please adjust the x-axis that the 800 nm is readable. Also, in line with the other figures, please write “Wavelength (nm)”. In figures 6 (c) and (d), please adjust the axes to the same ranges for better comparison.

3. Conclusions

The section 3 is directly followed by section 5; where is section 4? Please adjust for the conclusions.

4. Discussion of the results.

You show that the incorporation of 0.05 wt % into the hybrid perovskite solar cell gives the best power conversion efficiency, whereas upon higher wt % the efficiency drops again. Why is this? Could you add a sentence elaborating more on this topic? Also, could you imagine a synthesis route towards even larger grain sizes?

Author Response

June 11th, 2022

Prof. Carla Paduretu

Associate Editor, Nanomaterials

Subject: Responses to Referees' comments for our manuscript (Nanomaterials-1747131)

Dear Prof. Paduretu,

Thank you for sending us the revision request letter of 2th June concerning our manuscript “Highly Regioselective Synthesis of Bisadduct[C70] Additive toward the Enhanced Performance of Perovskite Solar Cells” (Manuscript number: 1747131).

We have read your letter and the Referees' comments carefully, and find that the Referees' comments are very constructive, helpful and positive overall. We followed the Referees' suggestions and performed the AFM characterization. Meanwhile, we revised our manuscript according to the referee’s suggestion. Below are our point by point answers to those comments of the Referees including the list of revisions of the manuscript (which are highlighted by giving the text a yellow background in the marked copy of the revised manuscript).

Review 1

The authors present their results on the successful synthesis of a C70 based additive to increase the power conversion efficiency of hybrid perovskite based solar cells. The investigation seems to be sound and the results and discussion are presented well. Apart from some smaller issues listed below I can support the publication in Nanomaterials.

  1. Language

The whole manuscript would benefit from some language editing. At the moment, some of the sentences are misleading, making it harder to understand the content of the manuscript. Just as an example, on page 1, line 24, please replace “as the effective strategies” by “as effective strategies”. Further down, line 27, I assume you meant “widely used as electron transport layer”? Again, further down on line 29, the sentence “whereas which have exhibited superior performance” makes no sense at the moment. There are more instances like these, so please have a careful look at the whole manuscript again.

Answer: We thank the Referee for the constructive suggestion. We polished the whole articles and corresponding revised words were labelled with yellow background.

  1. Figures

 In figure 4 (a), please adjust the x-axis that the 800 nm is readable. Also, in line with the other figures, please write “Wavelength (nm)”. In figures 6 (c) and (d), please adjust the axes to the same ranges for better comparison.

 Answer: We thank the Referee for the carefulness. According to the referee’s suggestion, we revised the corresponding figures.

  1. Conclusions

The section 3 is directly followed by section 5; where is section 4? Please adjust for the conclusions.

 Answer: We thank the Referee for the carefulness. According to the referee’s suggestion, we have adjusted the section number.

  1. Discussion of the results.

You show that the incorporation of 0.05 wt % into the hybrid perovskite solar cell gives the best power conversion efficiency, whereas upon higher wt % the efficiency drops again. Why is this? Could you add a sentence elaborating more on this topic? Also, could you imagine a synthesis route towards even larger grain sizes?

Answer: We thank the Referee for the constructive suggestion. The incorporation ratio of additive within perovskite film generally have an optimal value. The fullerene additives play the role of heterogeneous nucleation sites for perovskite crystallization (N. P. Padture, et al. Chem, 2018, 4, 1404- 1415; S. F. Yang et al. J. Mater. Chem. A, 2019, 7, 2754-2763). We added the corresponding sentence in discussion section as “This can be attribute to that the higher additive incorporation within perovskite perhaps results in excess nucleation sites, leading to the smaller grain size and more grain boundary formation.” The larger grain size of perovskite film is a synergistic result of the doped ratio and the interaction of fullerene derivatives with perovskite. Therefore, it is difficult to determine this results only from the viewpoint of synthesis route.

Reviewer 2 Report

The subject chosen by the authors in this manuscript is of high interest, both in fundamental science and in industrial applications. The work can be regarded as suitable for publication, based on its presentation and results, but there are some very important issues that needs focused attention.

The SEM details are lacking from the Experimental Section. The Conclusions Section is formatted differently, compared with main text.

-line 27-insert a blank between words /cells(PSCs)/; this should be applied also to line 51, line 85 and so on, throughout the text;

-line 95-synthesis starts with 62 mg and ends up with 44 g, please correct;

-line 112- Figure S1-there are no supplementary materials!

-line 149- there is no Table S1!

-line 224- Figure S4 is missing!

-line 258- CCDC (Cambridge Crystallographic Data Centre) number is missing!

All these missing (as no supplementary material is presented) need to be corrected. In this form the manuscript cannot be accepted, as it is misleading. After corrections and incorporation of the missing data it can be reconsidered for publication.

Author Response

June 11th, 2022

Prof. Carla Paduretu

Associate Editor, Nanomaterials

Subject: Responses to Referees' comments for our manuscript (Nanomaterials-1747131)

Dear Prof. Paduretu,

Thank you for sending us the revision request letter of 2th June concerning our manuscript “Highly Regioselective Synthesis of Bisadduct[C70] Additive toward the Enhanced Performance of Perovskite Solar Cells” (Manuscript number: 1747131).

We have read your letter and the Referees' comments carefully, and find that the Referees' comments are very constructive, helpful and positive overall. We followed the Referees' suggestions and performed the AFM characterization. Meanwhile, we revised our manuscript according to the referee’s suggestion. Below are our point by point answers to those comments of the Referees including the list of revisions of the manuscript (which are highlighted by giving the text a yellow background in the marked copy of the revised manuscript).

Review 2

Comments and Suggestions for Authors

The subject chosen by the authors in this manuscript is of high interest, both in fundamental science and in industrial applications. The work can be regarded as suitable for publication, based on its presentation and results, but there are some very important issues that needs focused attention.

Answer: We thank the Referee for his/her positive review.

The SEM details are lacking from the Experimental Section. The Conclusions Section is formatted differently, compared with main text.

Answer: We thank the Referee for the constructive suggestion and we have added the SEM details in the Experimental section as well as revised the format of Conclusion section.

-line 27-insert a blank between words /cells(PSCs)/; this should be applied also to line 51, line 85 and so on, throughout the text;

Answer: We thank the Referee for his/her carefulness and constructive suggestion. According to the referee’s suggestion, we rechecked the whole articles and revised these mistakes mentioned above.

-line 95-synthesis starts with 62 mg and ends up with 44 g, please correct;

Answer: We thank the Referee for his/her carefulness and constructive suggestion. According to the referee’s suggestion, we revised these expression mentioned above.

-line 112- Figure S1-there are no supplementary materials!

Answer: We thank the Referee for his/her carefulness and constructive suggestion. According to the referee’s suggestion, we revised these mistakes mentioned above.

-line 149- there is no Table S1!  -line 224- Figure S4 is missing!

Answer: We thank the Referee for his/her carefulness and constructive suggestion. According to the referee’s suggestion, we revised these mistakes mentioned above.

-line 258- CCDC (Cambridge Crystallographic Data Centre) number is missing!

Answer: We thank the Referee for his/her carefulness and constructive suggestion. The CCDC number have been contained in section “2.4 Crystal growth and measurements”.

All these missing (as no supplementary material is presented) need to be corrected. In this form the manuscript cannot be accepted, as it is misleading. After corrections and incorporation of the missing data it can be reconsidered for publication.

Answer: We thank the Referee for his/her carefulness and constructive suggestion. According to the referee’s suggestion, we rechecked the whole articles and revised these mistakes mentioned above.

Reviewer 3 Report

The paper entitled “Highly Regioselective Synthesis of Bisadduct[C70] Additive toward the Enhanced Performance of Perovskite Solar Cells” has been submitted by Qiu et al., in Nanomaterials. Please address the following comments.  

The improvement in the PCE was only 7% which is not sufficient to claim the superiority of this work. I recommend please provide a quantitative analysis to support the results. Novelty needs to be highlighted.

The authors focused on synthesizing C70 additive to enhance the performance of perovskite solar cells. PCBM has been widely used in PSCs as ETL as well as additive. Can the authors comment on the utilization of another fullerene derivative such as ICBA in perovskite?

What is the difference between control and 0.0wt% devices?

The analysis of the device performance is quite weak. For example, there is a need to discuss the effect of varying percentages (wt%) on the photovoltaic parameters (FF, VOC, and JSC). How does variable wt% influence these parameters?

Can the authors provide the surface analysis via AFM, because it is very important considering the improvement in the performance?

I suggest including the recent studies incorporating fullerene derivatives in the introduction for improving the quality. (https://doi.org/10.1016/j.jpowsour.2021.230782; https://doi.org/10.1016/j.jmrt.2021.12.086)

Author Response

June 11th, 2022

Prof. Carla Paduretu

Associate Editor, Nanomaterials

Subject: Responses to Referees' comments for our manuscript (Nanomaterials-1747131)

Dear Prof. Paduretu,

Thank you for sending us the revision request letter of 2th June concerning our manuscript “Highly Regioselective Synthesis of Bisadduct[C70] Additive toward the Enhanced Performance of Perovskite Solar Cells” (Manuscript number: 1747131).

We have read your letter and the Referees' comments carefully, and find that the Referees' comments are very constructive, helpful and positive overall. We followed the Referees' suggestions and performed the AFM characterization. Meanwhile, we revised our manuscript according to the referee’s suggestion. Below are our point by point answers to those comments of the Referees including the list of revisions of the manuscript (which are highlighted by giving the text a yellow background in the marked copy of the revised manuscript).

Review 3

Comments and Suggestions for Authors

The paper entitled “Highly Regioselective Synthesis of Bisadduct[C70] Additive toward the Enhanced Performance of Perovskite Solar Cells” has been submitted by Qiu et al., in Nanomaterials. Please address the following comments.  

The improvement in the PCE was only 7% which is not sufficient to claim the superiority of this work. I recommend please provide a quantitative analysis to support the results. Novelty needs to be highlighted.

Answer: We thank the Referee for his/her professional and constructive suggestion. We have revised the expression about the PSCs performance and added the contents to elucidate the underlying reason for the improve device performance. “Among the three photovoltaic parameters, the PCE enhancement after the incorporation of additive 2 primarily resulted from the increase of Jsc (from 22.15 to 22.20 mA cm-2, ~0.2 % enhancement), and FF (from 71.41% to 73.21%, ~2.5% enhancement).”

The authors focused on synthesizing C70 additive to enhance the performance of perovskite solar cells. PCBM has been widely used in PSCs as ETL as well as additive. Can the authors comment on the utilization of another fullerene derivative such as ICBA in perovskite?

Answer: We thank the Referee for the constructive suggestion. PCBM is the most widely used fullerene derivative applied in perovskite as ETL or additive so far. Surely, ICBA is another impressive fullerene derivative applied as acceptor materials in polymer solar cells. While it acted as additive or ETL within perovskite film and the corresponding photovoltaic performance is far inferior to that of the PCBM counterpart. (Adv. Mater., 2013, 25, 3727–3732; Adv. Energy Mater., 2015, 5, 1402321; Nanoscale, 2016, 8, 4077-4085).

What is the difference between control and 0.0wt% devices?

Answer: We thank the Referee for the constructive suggestion. The control devices is the standard device for contrast. While the 0.0wt% devices herein were used to preclude the influence of chlorobenzene solvent introduced in the procedure of device fabrication. We have elucidated this reason as “To preclude the influence of CB solvent on the device performance, we added the same amount volume of CB into perovskite layer to construct the devices (0 wt% additive).”

The analysis of the device performance is quite weak. For example, there is a need to discuss the effect of varying percentages (wt%) on the photovoltaic parameters (FF, VOC, and JSC). How does variable wt% influence these parameters?

Answer: We thank the Referee for the constructive suggestion. We have explained that varying doping ratio (wt%) of additive 2 with perovskite film change the number of heterogeneous nucleation sites, which regulate the crystallization rate and grain size as well as the corresponding trap state density. As the results, the photovoltaic parameters (FF, VOC, and JSC) would be affected by the varied doping ratio of 2.

Can the authors provide the surface analysis via AFM, because it is very important considering the improvement in the performance?

Answer: We thank the Referee for the constructive suggestion. We added the AFM characterization shown Figure S4. We also added the discussion “Roughness (25.17 nm) of perovskite films wiht additive 2 (0.05 wt%) is larger than that (12.45 nm) of the control perovskite film, which is attributed to the aggregation of fullerene derivative 2 similar to reported C60-PyP additive.[28]”

I suggest including the recent studies incorporating fullerene derivatives in the introduction for improving the quality. (https://doi.org/10.1016/j.jpowsour.2021.230782; https://doi.org/10.1016/j.jmrt.2021.12.086)

Answer: We thank the Referee for the constructive suggestion. We have cited the suggested articles as refer. 4 and 5.

Round 2

Reviewer 2 Report

The revised form of the manuscript is now suitable for publication, still, after some minor typos and corrections, that can be done under the proofing process. The missing Supplementary part was also now added, thus improving the manuscript.

Author Response

June 12th, 2022

Prof. Carla Paduretu

Associate Editor, Nanomaterials

Subject: Responses to Referees' comments for our manuscript (Nanomaterials-1747131)

Dear Prof. Paduretu,

Thank you for sending us the revision request letter of 2th June concerning our manuscript “Highly Regioselective Synthesis of Bisadduct[C70] Additive toward the Enhanced Performance of Perovskite Solar Cells” (Manuscript number: 1747131).

We have read your letter and the Referees' comments carefully, and find that the Referees' comments are very constructive, helpful and positive overall. According to the Referees' suggestions, we checked the spell mistakes and the cited references of whole manuscript. The revised parts in manuscript were labelled with a green background and contains the“Track Changes”.

Reviewer 3 Report

It can be accepted

Author Response

(The authors gave the same response as above.)
